



# Brief communication: On the definition of the low-level jet

Christoffer Hallgren[1,*], Jeanie A. Aird[2,*], Stefan Ivanell[1], Heiner Körnich[3], Rebecca J. Barthelmie[2], Sara C. Pryor[4], and Erik Sahlée[1]

[1]Department of Earth Sciences, Uppsala University, Uppsala, Sweden
[2]Sibley School of Mechanical and Aerospace Engineering, Cornell University, Ithaca, New York, USA
[3]Swedish Meteorological and Hydrological Institute, Norrköping, Sweden
[4]Department of Earth and Atmospheric Sciences, Cornell University, Ithaca, New York, USA

**Correspondence:** Christoffer Hallgren (christoffer.hallgren@geo.uu.se), Jeanie A. Aird (jaa377@cornell.edu)

**Abstract.** Low-level jets (LLJ) are examples of non-ideal wind speed profiles affecting wind turbine power production, wake recovery, and structural/aerodynamic loading. However, there is no consensus regarding which definition should be applied for jet identification. In this study we argue that a shear definition is more relevant for wind energy than a falloff definition. The shear definition is demonstrated and validated through development of an ERA5 LLJ climatology for six sites. Identification of LLJ and their morphology, frequency, and intensity is critically dependent on the i) vertical window of data from which LLJ are extracted and ii) the definition employed.

## 1 Introduction

The shape of the wind speed profile across the rotor has a pronounced impact on wind power generation. It affects the actual power production from a single turbine (Nunalee and Basu, 2014; Weide Luiz and Fiedler, 2022) as well as the wake properties and thus also the total production from a wind farm (Doosttalab et al., 2020; Gadde and Stevens, 2021). Further, the wind profile also dictates structural shear loads and mechanical stress across the rotor plane (Gutierrez et al., 2017, 2019). Usually, within the surface layer of the atmosphere, the wind profile exhibits an approximately logarithmic form as predicted by similarity theory (Motta et al., 2005). Even beyond the surface layer, there is an a priori expectation that winds will increase monotonically with increasing height. However, this is not always the case. In some areas, both offshore and onshore, non-ideal wind profiles (i.e., those that deviate from those inferred from Monin-Obukhov similarity theory) are common due to meteorological conditions manifest from mesoscale to the synoptic scale (i.e., day-night baroclinicity, complex topography and sloping terrain, coastal geography, and frontal passages).

One specific type of non-ideal profile displays a local maximum in the wind speed profile (a core) and is referred to as a low-level jet (LLJ) (illustrated in Fig. 1a). Although LLJ have been studied in both climatological and wind energy contexts worldwide (Algarra et al., 2019; Lima et al., 2022), there is no general consensus within the field regarding how to distinguish a LLJ from a non-LLJ wind speed profile (i.e., the jet definition) at heights relevant to wind energy. Most commonly, an absolute threshold of the minimum falloff (the decrease in wind speed above and below the core) is applied. Typically, this threshold is either 1 m s$^{-1}$ or 2 m s$^{-1}$ (e.g., Hallgren et al., 2020; Kalverla et al., 2020). Both Baas et al. (2009) and Aird et al. (2021)





argued that this absolute threshold should be combined with a relative threshold such that the wind speed above and below the
25 core also has to decrease by at least 10 or 20% of the core speed. Less commonly, the LLJ definition is dependent on thresholds
for negative shear above the wind speed profile core (e.g., Weide Luiz and Fiedler, 2022). Sometimes, an additional criterion of
the duration of an event is added as part of the identification of the LLJ (Svensson et al., 2019). The vertical extent of the wind
profile assessed for LLJ typically ranges up to 300 or 500 m, in some studies even higher. Inconsistencies in LLJ definitions
result in numerical uncertainties for ensemble LLJ characteristics (mean ensemble jet core speeds differ approximately 2 m s$^{-1}$
on average between definitions) rendering comparisons between studies difficult (Aird et al., 2021).

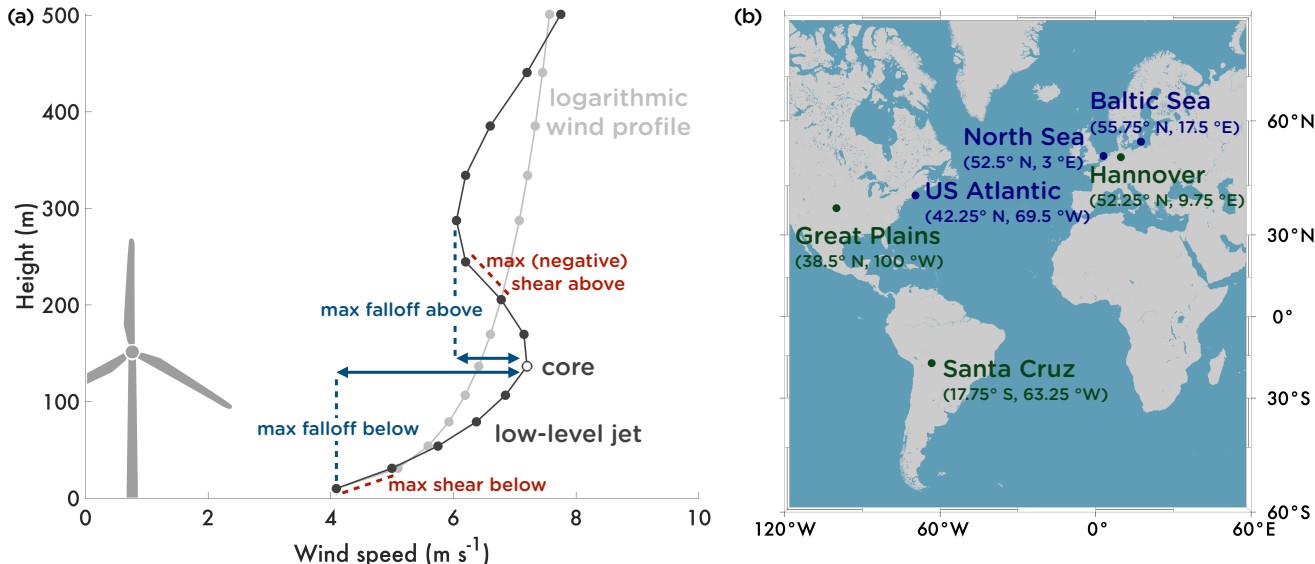

**Figure 1.** In panel (a), an example of an LLJ wind profile is shown, plotted with the ERA5 reanalysis vertical resolution (see Sect. 2 for
further details). For reference an ideal (logarithmic) wind speed profile is denoted in light grey. The maximum falloffs in wind speed above
and below the core are indicated in blue, and the strongest shears below and above the core are indicated in red. The map in panel (b) shows
the locations of the three offshore/coastal sites (blue) and the three onshore sites (green) analyzed in this study.

Compared to a falloff definition of LLJ, a shear definition considers the wind speed rate of change with height, rather than
only the change of wind speed independent of the height. Strong negative shear is associated with a low degree of turbulence,
enhanced entrainment fluxes (Doosttalab et al., 2020) and possibly a separation of atmospheric layers with different turbulent
properties (Banta et al., 2006; Hallgren et al., 2022), which in turn results in different effects on the loads on the wind turbine
and wake recovery rates (Doosttalab et al., 2020; Gadde and Stevens, 2021). For an extreme example, assume that the vertical
wind speed profile is discretized with an interval of 50 m. If the wind speed decreases by 0.99 m s$^{-1}$ between two levels (and
then slowly increases to the next level), the profile will not be classified as an LLJ according to a 1 m s$^{-1}$ falloff threshold,
although having a very strong shear of $-0.0198$ s$^{-1}$ in this layer. On the other extreme, if the wind speed decreases evenly by
1 m s$^{-1}$ between, e.g., 100 m and 500 m, the shear is only $-0.0025$ s$^{-1}$ in this layer, although the profile is classified as an
40 LLJ according to the falloff threshold.





In this work, we compare how the selection of LLJ definition affects different aspects of the climatology, focusing on metrics relevant for wind power applications. The vertical resolution for ERA5 wind speed profiles used in the study is approximately 30 m between vertical levels. The most common falloff definition is compared to a shear definition, and the definitions are applied over a height interval starting below the lowest height swept by the rotor and extend well beyond the top height of the rotor, to allow for identification of jet cores below, close to and above hub height (Gadde and Stevens, 2021). The LLJ can then be filtered based on core height depending on the aim of the study.

## 2  Method

To identify well-pronounced LLJ that are of particular relevance to wind energy, thresholds of falloff or shear should be applied both below and above the local maximum in the wind profile. In this study, the two LLJ definitions compared are:

- **Falloff:** a change in horizontal wind speed of both at least +/- 1 m s$^{-1}$ and +/- 10% of the core speed below/above the core

- **Shear:** a wind shear of at least +/- 0.01 s$^{-1}$ below/above the core

The definitions are applied in the same way for three offshore/coastal sites (the Baltic Sea, the North Sea/Southern Bight, the US Atlantic coast east of Boston) and three onshore sites (the Great Plains/Western Kansas, Hannover, Santa Cruz) using data from the European Centre for Medium-Range Weather Forecasts (ECMWF) fifth generation reanalysis (ERA5) (Hersbach et al., 2020). Hourly output from the period 1979–2022 is analyzed. Although wind speed profiles from ERA5 generally exhibit pronounced smoothing when compared to lidar measurements (Hallgren et al., 2020), the study herein is methods-focused and aims to compare the two methods of LLJ extraction, and the subsequent effect on LLJ characterization, rather than developing a robust LLJ climatology. As such, the same definitions for LLJ identification have been applied both onshore and offshore, although it could be argued that different criteria should be applied in different locations. However, the scope of this study is to investigate and compare the qualitative and quantitative LLJ characteristics both offshore and onshore when implementing the shear and falloff definitions. Subsequently, this study does not vary the quantitative criteria, but future studies could build upon this work by optimizing and refining the criteria offshore and onshore for different wind energy applications. The sites represent regions relevant for wind power development, where LLJ have previously been studied and the locations of the selected grid points are presented in Fig. 1b.

As the rotor plane typically covers the height range 30 – 300 m (Barthelmie et al., 2020), we suggest to apply the definition over 10–500 m to stretch well beyond the rotor plane, when data are available. In our comparison, the ERA5 wind speed profile from model level numbers 137–124, corresponding to 10–500 m above the surface in the standard atmosphere, is assessed for LLJ. The approximate height of the levels are indicated in Fig. 1a. Absolute heights above ground of the model levels are calculated relative to each time step and coordinate location as a function of thermal and humidity conditions within the ERA5 vertical profile. To estimate the power production during hours with LLJ extracted by each definition, the rotor equivalent wind speed (see e.g., Barthelmie et al., 2020) is calculated and applied to a power curve, assuming the 15 MW reference turbine





(hub height: 150 m, radius: 120 m) described by Gaertner et al. (2020) for the three offshore sites and a Vestas 150-4.2 MW (hub height: 155 m, radius: 75 m) for the three onshore sites (Vestas, 2023).

## 3 Results and discussion

The distributions of absolute and relative falloff as well as the strongest shear above and below the core for all wind profiles with a local maximum are presented for two of the sites (the Baltic Sea and the Great Plains) in Fig. 2. Qualitatively, the results are similar for all of the three offshore sites and for all of the three onshore sites. The distributions of all profiles with a local maximum in the wind profile (yellow in Fig. 2) indicate that, although the wind profiles are all non-ideal and have

a local maximum, most of the time the core is not that pronounced (i.e. the shear around the core is relatively low). Due to frictional dissipation of momentum at the surface, both the increase of the wind speed and the shear below the core are generally stronger than the corresponding values above the core. Comparing the lines indicating the thresholds for the falloff and the shear definition of LLJ with the distributions, it is clear that the definitions overlap to a large extent, but not fully. More LLJ are identified with the falloff definition, within the range of approximately 50–150% more for all sites. The overlap is

presented in the Venn diagrams in Fig. 2d. For the Baltic Sea, 25% of the time steps identified as exhibiting LLJ found by the shear definition are unique to that definition, but there is an overlap of 75%. The falloff definition systematically identifies more LLJ by 49–67% across the six sites (See Fig 3. for examples of temporal mean LLJ profiles extracted with each definition).

To compare the effect of the selection of the definition, statistics for commonly studied LLJ attributes are presented for all sites in Fig. 3. Despite the fact that both definitions partly identify unique LLJ, the mean profiles are qualitatively similar (Fig.

3a) in shape. On average the LLJ occur in weaker wind conditions than the climatology, especially onshore, while offshore the LLJ identified with the shear definition occur at comparable or somewhat higher wind speeds. Core height (Fig. 3b) and speed (Fig. 3c) statistics extracted by each definition are similar, but the dispersion in core height is higher and skewed towards higher altitudes for the shear definition (6–10% higher on average, except for US Atlantic with 1% lower average core height). The height of the core is related to the wake recovery rate, with higher cores resulting in shorter wakes, as discussed by Gadde

and Stevens (2021). Related to the height of the core and the positive shift in the average wind profile, the average core speed is 7–22% larger for LLJ extracted by the shear definition (lowest values for US Atlantic).

Offshore, the distributions of power production under LLJ extracted with the shear definition are more similar to the climatology than those identified with the falloff definition, while onshore the power production is generally lower for both types of LLJ comparing to the climatology (Fig. 3d). Generally for both definitions, 97–99% of the offshore LLJ occur when the

turbine is generating power (rotor equivalent wind speed between cut-in and cut-out); onshore the numbers are only 4.2–20% for Hannover and Santa Cruz, but for Great Plains approximately 50–60%. Comparing with the climatology, the turbine is generating power 90–92% of the time offshore and 41–69% of the time onshore (highest values for the Great Plains). Onshore, LLJ identified by the falloff definition are more likely to appear at times when the turbine is not generating power, compared to those LLJ found by the shear definition. Offshore, differences between definitions in that regard are small, as indicated by

the percentage range.





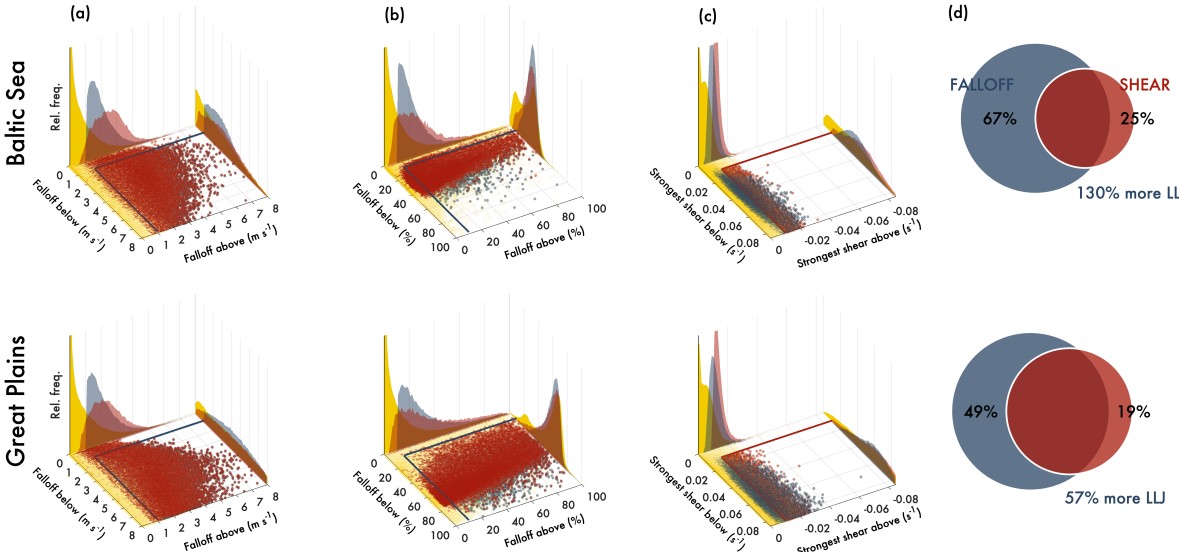

**Figure 2.** Distributions of all profiles with a local maximum in the wind profile (yellow) in terms of absolute (column a) and relative (column b) change of wind speed above and below the core and strongest shear above and below the core (column c) for the Baltic Sea (top row) and the Great Plains (bottom row). The combined +/- 1 m s$^{-1}$ and 10% falloff criterion is marked with the dark blue line (in column a and column b) and the identified LLJ as pale blue dots. Similarly, the +/- 0.01 s$^{-1}$ shear criterion is marked with the red line (in column c) and the identified LLJ as red dots. Distributions of the relative frequency for all profiles with a local maximum (yellow), LLJ identified by the falloff definition (blue) and by the shear definition (red) are projected onto the vertical planes. In column (d), Venn diagrams show the overlap between the two LLJ definitions. The size of the circles is proportional to the number of identified LLJ. The percentages in black indicate the amount of unique identifications for each definition.

The seasonality of LLJ extracted with both definitions are qualitatively similar (Fig. 3e) for most sites, although a higher frequency of LLJ is found with the falloff definition (see also Fig. 2d). There is a clear peak in LLJ frequency during the summer for offshore sites and a less pronounced peak in (hemispheric) winter for onshore sites. However, for Santa Cruz, LLJ extracted with the falloff definition do not exhibit seasonal variability. For the onshore sites, there is evidence of a diurnal cycle

(Fig. 3f), which is most pronounced for the Great Plains, with a peak in LLJ frequency around 6 LST.

Using the shear definition, more shear-symmetric LLJ are found for all sites (Fig. 3h) and there is a markedly higher dispersion of the ratio of the maximum falloff above/below the core (Fig. 3g). This finding is particularly relevant for wind energy. As such, the shear definition identifies LLJ with more variability in the ratio of the maximum falloff above/below the core, which has implications for aerodynamic and structural loading (Gutierrez et al., 2019) as well as wake recovery (Gadde and

115 Stevens, 2021). This is a key differentiating factor when compared to LLJ extracted by the falloff definition, and gives a strong indication that the shear definition is superior to the falloff definition; the shear definition is identifying more LLJ with unique wind profiles that are relevant to wind turbine power production and lifespan.





The persistence plot (Fig. 3i) shows the duration of LLJ events. Most LLJ appear as single-hour events, with approximately 20% for LLJ found by the falloff definition and 30–40% of the LLJ found by the shear definition. There is a gradual decrease in the probability of events with longer duration. The red noise component of atmospheric behavior (i.e., the persistence) means that the probability of an LLJ with a given duration for lags of >1 hour exceeds that derived using a random number generator that has the correct joint probability by hour of the day and month of the year. In Fig. 3j, the difference in the distributions of the persistence of LLJ events is plotted, comparing the duration of LLJ events with what could be expected from a randomized 100 year climatology, keeping the seasonal and diurnal relative frequencies as found by either the falloff or the shear definition for each site, respectively. Compared to the red noise, LLJ tend to cluster in events. Singular LLJ appear 50–60 percentage points less often than what would be expected from the random climatology. Events lasting two hours are more common using the shear definition, but in general both definitions are similar in the persistence of LLJ events, when taking the differences in relative occurrence between the two definitions into account (Fig. 3j).

In summary, the shear-based LLJ definition results in qualitatively similar LLJ characteristics in terms of average profiles, power production and diurnal/seasonal pattern (Fig. 3) when compared to the falloff definition, although the definitions partially identify unique LLJ (Fig. 2d). This may be attributed to each definition identifying LLJ in different atmospheric regimes, i.e. it is possible the shear definition identifies a higher fraction of LLJ in stable conditions due to the lower mean core speeds exhibited by shear-extracted LLJ. However, as the shear definition considers the change in wind speed over the change in height, it is better suited for wind power applications due to the relationship between change in speed over height and bending moment calculations, a primary consideration of LLJ occurrence (Gutierrez et al., 2017; Gadde and Stevens, 2021; Weide Luiz and Fiedler, 2022).

The falloff definition is highly sensitive to the vertical window applied, identifying 300-1200% more jets when increasing the vertical window from 10–300 m to 10–500 m (results not shown). The shear definition is less sensitive, only identifying 80–300% more jets in the larger vertical window. This may be attributed to frequent reductions in wind speed with height when the vertical window is extended, that allow for fulfillment of the fall-off definition but do not exhibit strong enough intra-layer shear to meet the shear definition. LLJ definitions based on e.g., convergence of momentum flux could also be suitable for wind power, but suffer from measurement difficulty in practical application. However, the shear definition proposed in this study relies solely on measurements from the wind speed profile and could thus easily be applied to model or measurement data. Using multi-year or multi-decade data (as in this study), a statistical definition could be created to classify the most extreme profiles in terms of falloff or shear as LLJ. This would result in a more site specific definition (which would make inter-study comparisons more difficult) but could, on the other hand, provide a more robust definition that is not based on an arbitrary threshold (such as $1 \text{ m s}^{-1}$, 10%, or $0.01 \text{ s}^{-1}$).

## 4 Conclusions

LLJ climatologies at three offshore and three onshore sites are developed using two different LLJ definitions – one (traditional) falloff-based definition and one (novel) shear-based definition. The LLJ climatologies generated by each definition are com-

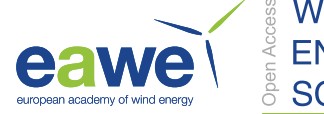

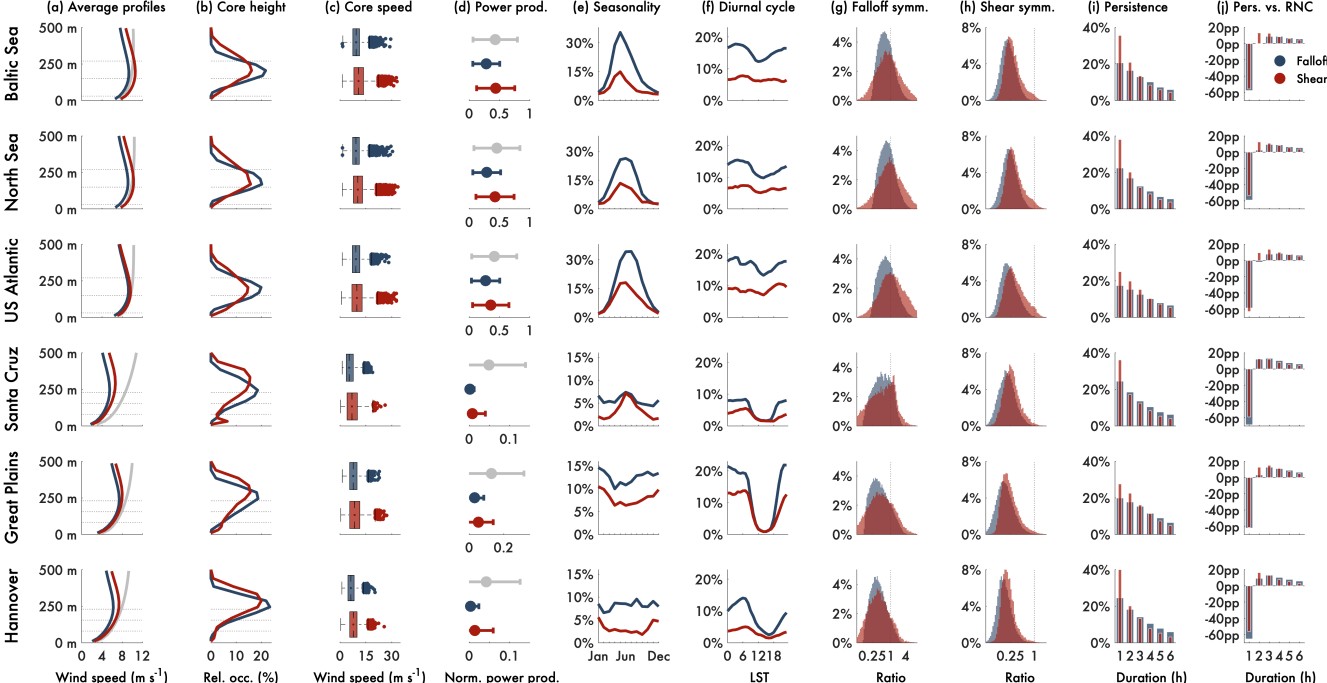

**Figure 3.** Statistics of LLJ characteristics for the six sites using 44 years of hourly ERA5 data (1979–2022). In column (a) the average wind speed profiles are plotted for the two LLJ definitions, together with the climatological wind speed profile at the site (in grey). Distributions of LLJ core height and core speed are shown in columns (b) and (c). In columns (a) and (b) the hub height and the top and bottom heights swept by the turbine blades are marked with dotted lines (see Sect. 2 for turbine types). The mean power production (normalized with the rated power) and the standard deviation of the mean for LLJ found by the two definitions and the climatology (in grey) is plotted in column (d). In column (e) the relative occurrence of LLJ throughout the year is plotted and in column (f) the diurnal cycle in local sidereal time (LST) is shown. Note the different scales on the ordinate for offshore (top three rows) and onshore (bottom three rows) locations. In column (g), the symmetry of the LLJ in terms of the ratio of the falloff above/below the core is shown and in column (h) the ratio of the absolute value of the strongest shear above the core to the strongest shear below the core. In column (i) the duration of LLJ events is plotted normalized against the total number of events. In column (j) the change in percentage points of the duration of LLJ events compared to a red noise climatology (RNC), following the same seasonal and diurnal relative occurrence as in columns (e) and (f). In columns (a), (e) and (f) the 99.9% confidence interval of the mean hardly differs from the line thickness, and thus, is not shown. For the boxplots in column (c) and (d), the dot in each box marks the median value and notches in the box mark the 95% confidence interval of the median. Edges of the box show 25th and 75th percentiles respectively and whiskers extend from the box to the most extreme value within 1.5 times the interquartile range. Colored dots outside the whiskers mark the outliers.

pared. In general the two definitions give similar results for all sites studied, with differences mainly associated with the total frequency of occurrence (50–150% more LLJ using the falloff definition), wind speed (a shift towards higher wind speeds in the average profiles and for the core speed for the shear definition), core height (somewhat greater variability and in general





6–10% higher average core height for the shear definition) and symmetry (more shear-symmetric LLJ with a larger variability
in falloff ratio above/below found by the shear definition).

  Based on the results, we recommend the shear definition as the optimal LLJ definition for wind energy applications. The
shear definition better captures sharp transitions in the wind profile (as evidenced by the higher dispersion in falloff ratios
exhibited by LLJ extracted with the shear definition, see Fig. 3g), which are decisive for structural and aerodynamic loading
as well as wake recovery rates (Gutierrez et al., 2017, 2019; Doosttalab et al., 2020; Gadde and Stevens, 2021). These sharp
transitions are possibly related to layers in the atmospheric boundary-layer with different properties (Sect. 1). Further, the shear
definition is less sensitive to the vertical window applied which is one of the primary sensitivities of LLJ characterization. The
shear threshold should be applied both below and above the local wind speed maximum in the vertical profile and preferably
over a height range extending beyond the heights swept by average offshore/onshore wind turbines to allow for identification
of jet cores below, close to, and above hub height.

*Code and data availability.* The code used to generate the figures can be acquired by contacting the corresponding author. All ERA5 data
(hourly values on model levels for wind components, temperature, and specific humidity, hourly data on a single level for surface pressure)
can be downloaded via the Copernicus Climate Change Service (C3S).

*Author contributions.* The conceptualization, administration, methodology, programming, validation, formal analysis, visualization and writ-
ing the original draft was performed by CH and JAA. CH was supervised by ES, SI and HK. JAA was supervised by RJB and SCP. Funding
acquisition was carried out by ES, SI, HK, RJB and SCP. All authors participated in reviewing and editing the manuscript.

*Competing interests.* At least one of the (co-)authors is a member of the editorial board of Wind Energy Science. Apart from that, the authors
declare no conflict of interest. The funding agencies had no role in the design of the study; in the collection, analyses, or interpretation of
data; in the writing of the manuscript, or in the decision to publish the results.

*Financial support.* This research was funded by the Energimyndigheten (Swedish Energy Agency) VindEl program, Grant Number 47054-1.
The work forms part of the Swedish strategic research program StandUp for Wind.





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
