# Peer review of "Brief communication: On the definition of the low-level jet"

_Wind Energy Science, 2023_

## Referee Comment (RC1)

**Brief communication: On the definition of the low-level jet**
*Hallgren et al.*

**REVIEW**

1. Why not combining both definitions for LLJ identification, as for example done in the Debnath et al. 2021 WES paper? (Anyways, I would also suggest adding that paper as a reference).
2. Also, can you analyze the sensitivity of the single definitions to the thresholds you select for each? I expect some of the differences to increase/decrease if you use more/less strict thresholds.
3. L.19: should be "LLJs" (the same applies in several other places in the draft)
4. Mention somewhere that the jet core is also sometimes referred to as jet nose.
5. L.50: specify, for both definitions, if "above/below" means "between above and below" or "between above and the core, AND below and the core" or "between above and the core, OR below and the core".
6. L.80: comma after "i.e." (the same applies elsewhere too)
7. Figure 2: I find this figure rather challenging to understand given the small size of the plots and overlap in the points. How about not including the points on the horizontal planes at all, and just show the histograms and Venn diagrams? With this scenario, I would also suggest having the yellow, blue, and red histograms all share the same y-axis, so that the reader can get a sense of the frequency of LLJs compared to all profiles with a local maximum. Also, the percentages in panel (d) are not clear. Does it mean, for example, that 67% of the yellow cases are classified as LLJs in the Baltic Sea, when using the falloff definition? It wouldn't seem like that's the case from the histograms as they are shown now.
8. Can you include the (revised) plots for all sites in a supplement?
9. L.110: specify AM or PM.
10. L.150: the shear-based definition is not really novel.
11. L.165: I believe you need to add a reference entry for the data used, according to Copernicus' policies.

---

## Author Comment (AC1)

**Author's response**

Dear Referee #1,

Thank you for your feedback that helped us further improve and strengthen our manuscript. An overview of the changes in the manuscript related to your comments are presented below.

**1. Why not combining both definitions for LLJ identification, as for example done in the Debnath et al. 2021 WES paper? (Anyways, I would also suggest adding that paper as a reference).**

Thank you for this comment and excellent suggestion of work to be referenced. We are aware of the Wind Energy Science guideline of a maximum of 20 references in a Brief communication, but we hope that the editor is okay with us adding this extra reference.

The reason we are not combining the two definitions is to be able to clearly differentiate the effect on the results on the selection of the definition being employed. However, although we are not presenting results explicitly for the combination of the two definitions, Fig. 2 indicates what would be the effect of such a combined definition. This is shown both in terms of which individual profiles that would be excluded (see e.g., the red dots outside of the blue line in panel b, showing that the shear definition identifies some profiles as LLJs although the relative falloff below the core is not reaching the threshold applied by the falloff definition) and the overlap between definitions (indicated by the Venn diagrams (panel d) in the old version of the manuscript, and on lines 87–90 in the revised version).

We agree that it is an interesting idea to combine the definitions, and in the revised draft we have added this as a suggestion for future work in the Discussion section. Lines 144–145 now read:

*To build on this study, the effect of using a combination of the falloff and shear definitions could be investigated (similar to what was used by Debnath et al. 2021)*

**2. Also, can you analyze the sensitivity of the single definitions to the thresholds you select for each? I expect some of the differences to increase/decrease if you use more/less strict thresholds.**

This could easily be done, and results for different combinations of absolute and relative falloff criteria applied both on model data and lidar observations of the wind speed profile are presented in earlier work by us (Aird et al. 2021; Hallgren et al. 2022). However, as the aim of our Brief communication is to compare the two qualitatively different definitions – rather than comparing the results for different thresholds employed by a single type of definition – we relinquish from this task. As is also mentioned in the manuscript, we suggest that statistical methods could be used to find the optimal shear threshold for LLJ identification for a specific site if multi-decade data is available, refraining from using arbitrary thresholds (such as e.g., 0.01 s$^{-1}$), see lines 148–151 in the revised manuscript.

**3. L.19: should be "LLJs" (the same applies in several other places in the draft)**

We have changed accordingly.

**4. Mention somewhere that the jet core is also sometimes referred to as jet nose.**

This is added on lines 18–19.

**5. L.50: specify, for both definitions, if "above/below" means "between above and below" or "between above and the core, AND below and the core" or "between above and the core, OR below and the core".**

This is an important clarification. Thank you for this suggestion. In the revised manuscript, the definitions on lines 51–54 now reads:

- *Falloff: an increase in horizontal wind speed of at least 1 m s$^{-1}$ and 10% of the core speed below the core and simultaneously a decrease of 1 m s$^{-1}$ and 10% above the core*

- **Shear:** *a local wind shear (i.e., between two vertical levels, see Fig. 1a) of at least 0.01 s$^{-1}$ below the core and simultaneously at least -0.01 s$^{-1}$ above the core*

**6. L.80: comma after "i.e." (the same applies elsewhere too)**

This has been corrected.

**7. Figure 2: I find this figure rather challenging to understand given the small size of the plots and overlap in the points. How about not including the points on the horizontal planes at all, and just show the histograms and Venn diagrams? With this scenario, I would also suggest having the yellow, blue, and red histograms all share the same y-axis, so that the reader can get a sense of the frequency of LLJs compared to all profiles with a local maximum. Also, the percentages in panel (d) are not clear. Does it mean, for example, that 67% of the yellow cases are classified as LLJs in the Baltic Sea, when using the falloff definition? It wouldn't seem like that's the case from the histograms as they are shown now.**

We agree that Fig. 2 is challenging and it takes some time to get used to the plots. There is a lot of information we want to convey in a compressed format, and reconsidering the plot and alternative ways of plotting we still conclude that this is probably the clearest way of presenting all the information. However, we enlarged panels (a)–(c) so that the points are easier to distinguish.

Keeping the data points on the horizontal plane is important because it provides information about the relationship between the two distributions shown in the vertical planes. While it is true that – if the variables were independent – the point clouds could be omitted, this is not the case here since there is a strong correlation between e.g., the wind speed below and above the local maximum. Further, the scale on the ordinate is not written out as it is dependent on the bin size used to create the histograms. However, the bin size is of course the same for all histograms within a panel, and thus the histograms in each panel are comparable.

You are absolutely right that the percentages given in the Venn diagrams are not clear. What we meant to say is that, e.g., for the Baltic Sea 67% of the LLJs identified by the falloff definition were unique to that definition and that 25% of the LLJs identified by the shear definition were unique to that definition. In other words, 33% of the LLJs found by the falloff definition were also found by the shear definition, and 75% of the LLJs found by the shear definition were also found by falloff definition. This has been clarified in the revised manuscript. Based on your comment, we also decided to remove the Venn diagrams from Fig. 2, as these results easily could be described in text.

**8. Can you include the (revised) plots for all sites in a supplement?**

We have added a supplement where the revised plots for all sites can be found

**9. L.110: specify AM or PM.**

We have rephrased and clarified this sentence, which now reads (lines 112–113):

*For the onshore sites, there is evidence of a diurnal cycle (Fig. 3f), which is most pronounced for the Great Plains, with a peak in LLJ frequency during the night.*

**10. L.150: the shear-based definition is not really novel.**

We have removed the words "traditional" and "novel" describing the two definitions.

**11. L.165: I believe you need to add a reference entry for the data used, according to Copernicus' policies.**

Thank you for this comment. The data availability now reads (lines 175–179):

*For the ERA5 data (hourly values on model levels for wind components, temperature, and specific humidity, hourly data on a single level for surface pressure) we refer to Hersbach et al. (2017). Data were downloaded*

*from the Copernicus Climate Change Service (C3S) (2023). The results contain modified Copernicus Climate Change Service information. Neither the European Commission nor ECMWF is responsible for any use that may be made of the Copernicus information or data it contains.*

Once again, thank you for your comments.

Sincerely,
C. Hallgren, J.A. Aird and co-authors

**References**

J. A. Aird, R. J. Barthelmie, T. J. Shepherd, and S. C. Pryor. WRF-simulated low-level jets over Iowa: characterization and sensitivity studies. *Wind Energy Science*, 6(4):1015–1030, 2021. doi: 10.5194/wes-6-1015-2021.

Copernicus Climate Change Service (C3S). Complete era5 global atmospheric reanalysis, 2023. URL https://cds.climate.copernicus.eu/cdsapp#!/dataset/reanalysis-era5-complete. last access: 2023-03-31.

M. Debnath, P. Doubrawa, M. Optis, P. Hawbecker, and N. Bodini. Extreme wind shear events in us offshore wind energy areas and the role of induced stratification. *Wind Energy Science*, 6(4):1043–1059, 2021. doi: 10.5194/wes-6-1043-2021.

C. Hallgren, J. Arnqvist, E. Nilsson, S. Ivanell, M. Shapkalijevski, A. Thomasson, H. Pettersson, and E. Sahlée. Classification and properties of non-idealized coastal wind profiles–an observational study. *Wind Energy Science*, 7(3):1183–1207, 2022. doi: 10.5194/wes-7-1183-2022.

H. Hersbach, B. Bell, P. Berrisford, S. Hirahara, A. Horányi, J. Muñoz-Sabater, J. Nicolas, C. Peubey, R. Radu, D. Schepers, A. Simmons, C. Soci, S. Abdalla, X. Abellan, G. Balsamo, P. Bechtold, G. Biavati, J. Bidlot, M. Bonavita, G. De Chiara, P. Dahlgren, D. Dee, M. Diamantakis, R. Dragani, J. Flemming, R. Forbes, M. Fuentes, A. Geer, L. Haimberger, S. Healy, R.J. Hogan, E. Hólm, M. Janisková, S. Keeley, P. Laloyaux, P. Lopez, C. Lupu, G. Radnoti, P. de Rosnay, I. Rozum, F. Vamborg, S. Villaume, and J-N. Thépaut. Complete ERA5 from 1940: Fifth generation of ECMWF atmospheric reanalyses of the global climate. Copernicus Climate Change Service (C3S) Data Store (CDS), 2017. last access: 2023-03-31.

---

## Author Comment (AC2)

**Author's response**

Dear Referee #2,

Thank you for recognizing the relevance of our manuscript, your valuable feedback and constructive comments! We have revised the manuscript based on your comments and all changes made are presented below.

**The article presents a suggestion for a definition of the low-level jet (LLJ) with the aim to harmonize and make LLJ statistics of different sites easier to compare. The article compares statistics of the new methodology, which is based on wind shear below and above the jet core, with statistics based on the frequently used definition, which uses a threshold for the reduction of wind speed above the jet core. The results are compared for 3 offshore and 3 onshore sites of particular interest for wind energy.**

**The article addresses an important topic of making LLJ statistics comparable for different sites and methods. The new method is well justified and explained, and there are interesting results comparing the different sites. The paper is of large interest to the Wind Energy Community.**

**However, my main concern is that the new definition is only applied to simulation data. Such statistics is often determined from observational data, e.g. based on wind lidar profiles. If this method should be used by a broad community in the future, it is of high importance to take into account measurement data and compare to the statistics based on former criteria. Measurement data are usually not as "smooth" as simulation data, and there may be more variability of wind speed within one profile of wind speed. Also the treatment of lacking data from a certain height on, e.g. due to clouds, should be considered. If this provides similar results to the numerical data, and if the articles gives guidelines how to identify the LLJ form experimental data, e.g. in which height intervals, it will be much more convincing to the wind energy community.**

**The sites have been chosen at locations where observational data are available. So it requires large extra work, but should be possible, in principle. Therefore I would suggest to vote for "Major revision" before the article is ready for publication.**

Thank you for this comment. When we started this project we were considering to use lidar observations in the analysis but decided that a comparison of the two definitions benefit from a global dataset that is fully comparable between different sites and where multiple decades of data assures for robust statistics for rare events (i.e., LLJs). We agree that the next step in taking our analysis further is to perform the same analysis also on lidar measurements, and we have added a comment about this in the Discussion of the revised manuscript. Lines 152–157 now read:

*Building on this study, similar analysis on the best definition of the LLJ should be performed on wind speed profiles measured by lidar instruments. As modeled wind speed profiles often are much smoother than observed profiles (e.g., Hallgren et al. 2020; Kalverla et al. 2020) reanalyses tend to underestimate the actual occurrence of LLJs. Although difficult to compare sites with different lidar instrumentation (implying different vertical and temporal resolution, height range, and temporal coverage) this type of analysis would be of high importance for the wind energy industry, and results could be compared to those presented herein.*

As you mention, using observational data raises a lot more issues about data handling and rigorous quality control is needed, especially when analysing extreme events. Further, differences between instruments regarding vertical resolution, height range and uncertainties in the measurements as well as shorter time series (including gaps) requires more advanced statistical treatment to account for uncertainties in LLJ characteristics. Also, some sensitivity tests, such as sensitivity to vertical window, are not as straight forward using actual observations. That being said, we of course acknowledge that lidars measure the wind profile that a nearby wind turbine actually is exposed to, and thus – in many cases – is a data source of higher significance than model data. For now, we have to refer to validations of ERA5 compared to lidar measurements in cases of LLJs, e.g., Kalverla et al. (2019, 2020); Hallgren et al. (2020); Borvarán et al. (2021); Dieudonné et al. (2022); Rubio et al. (2022)

Keeping the manuscript in the compressed format of a Brief communication according to the Wind Energy Science guidelines, it is not feasible for us to include lidar data in our analysis as it would require an extensive expansion of the manuscript. We hope that both you and the editor find our work still worth of publishing as it provides results that will serve for reference in following studies where lidar data is analysed.

**Some remarks are listed below.**

**1: I would suggest to avoid using the term "non-ideal wind speed profiles" in the abstract. It should first be defined what you mean with this. Maybe substitute with "logarithmic"**

We changed to *non-logarithmic* in the abstract.

**17: Please provide references for the different phenomena that are associated with LLJ**

Following the Wind Energy Science guidelines for Brief communications, we are limited to a maximum of 20 references, and thus we had to make hard restrictions on which references were most important to include. However, we kindly ask the editor for permission to exceed this limit and have included the reference to Stensrud (1996) in our revised version of the manuscript. In this paper, the mechanisms for LLJ formation are summarized. For more detailed descriptions, we refer to some of the original papers on LLJ formation, such as e.g., Blackadar (1957) (diurnal variations in boundary-layer friction), Fisher (1960) (the sea breeze), Holton (1967) (diurnal variations in baroclinicity over sloping terrain), Kotroni and Lagouvardos (1993) (cold fronts), and Smedman et al. (1996) (coastal advection).

**31-34: The reason for the strong negative shear, low turbulence, entrainment and separation of layers is normally a temperature inversion. Maybe state this explicitly?**

Thank you for this comment. We have now clarified that these effects often are associated with a temperature inversion (lines 34–35 in the revised manuscript).

**35-39: Please compare the "extreme examples" to values of LLJ wind shear in the literature. E.g. for Emeis (2010) the wind shear values during LLJ events that were observed were only between around 0.03 and 0.095. So what you call "very strong shear" is below the lowest shear of experimental data.**

We have rephrased the examples and added a reference to Emeis (2014) for comparison. In the revised manuscript, lines 37–42 now read:

*For example, assume that the vertical wind speed profile is discretized with an interval of 30 m. If the wind speed decreases by 0.99 m s$^{-1}$ between two levels (and then slowly increases to the next level), the profile will not be classified as an LLJ according to a 1 m s$^{-1}$ falloff threshold, although having a shear of $-0.033$ s$^{-1}$ in this layer (compare to e.g., Emeis 2014). On the other hand, if the wind speed decreases evenly by 1 m s$^{-1}$ between, e.g., 100 m and 500 m, the shear is only $-0.0025$ s$^{-1}$ in this layer, although the profile is classified as an LLJ according to the falloff threshold.*

**55: Your results are based on ERA5. How well is ERA5 able to represent LLJ? Maybe there is a validation in the literature? If yes, please quote as a reference, if not, could you provide such an example yourself? Maybe at least exemplary for one month for one of the sites?**

Please see our reply to your main comment, where we have included references to validations of ERA5, comparing the reanalysis with lidar measurements, focusing on LLJs. All studies conclude that ERA5 (just like the other reanalyses) struggles resolving LLJ characteristics. Regional reanalyses (see e.g., Hallgren et al. 2020; Kalverla et al. 2020) might perform better in this respect, but we needed a global reanalysis for the purposes of this study to assure a fair comparison between sites. References to the aforementioned papers are included in the revised manuscript, see line 60 and lines 153–154.

**67: You suggest to make use of the height interval 10–500 m. For experimental data it would be good to have a suggestion how to calculate shear if this altitude range is not available. E.g. some lidars have an overlap issue and start measuring only from 40 m on, some have an upper range of 200m, etc.**

The calculation of shear is independent of the height interval and only depends on the height resolution. In other words, we are not considering the average shear in the full profile but rather all shear values in the profile. This is one of the reasons that the shear definition is less sensitive to the height interval and also means that the method to calculate the shear is directly applicable to lidar data covering different height intervals. In the revised manuscript, we have clarified that it is the *local* wind shear that is considered for the shear definition (see excerpt of the definitions below, or lines 51–54 in the revised manuscript).

**90 Unclear, please rephrase: "On average the LLJ occur in weaker wind conditions than the climatology"**

We have rephrased as *On average the LLJs occur in weaker wind speeds compared to the climatological mean*

**What about strong logarithmic increase of wind shear? Could this falsely be identified as a LLJ, e.g. in experimental data, when the wind speed minimum above the jet core is not covered by the data?**

No, it could not be falsely identified as an LLJ by the definitions we provide, as the wind shear has to reach certain thresholds both below and above the local maximum. So if the shear threshold above the local maximum is not met, then the profile is simply not classified as an LLJ.

In the revised manuscript, we have clarified both definitions (lines 51–54):

- ***Falloff:*** *an increase in horizontal wind speed of at least 1 m s$^{-1}$ and 10% of the core speed below the core and simultaneously a decrease of 1 m s$^{-1}$ and 10% above the core*

- ***Shear:*** *a local wind shear (i.e., between two vertical levels, see Fig. 1a) of at least 0.01 s$^{-1}$ below the core and simultaneously at least -0.01 s$^{-1}$ above the core*

Once again, thank you for all your comments.

Sincerely,
C. Hallgren, J.A. Aird and co-authors

**References**

A. K. Blackadar. Boundary layer wind maxima and their significance for the growth of nocturnal inversions. *Bulletin of the American Meteorological Society*, 38(5):283–290, 1957. doi: 10.1175/1520-0477-38.5.283.

D. Borvarán, A. Peña, and R. Gandoin. Characterization of offshore vertical wind shear conditions in Southern New England. *Wind Energy*, 24(5):465–480, 2021. doi: 10.1002/we.2583.

E. Dieudonné, H. Delbarre, A. Sokolov, F. Ebojie, P. Augustin, and M. Fourmentin. Characteristics of the Low-Level Jets Observed over Dunkerque (North Sea French coast) using 4 years of wind lidar data. *Quarterly Journal of the Royal Meteorological Society*, 2022. doi: 10.1002/qj.4480.

S. Emeis. Wind speed and shear associated with low-level jets over Northern Germany. *Meteorologische Zeitschrift*, 23(3):295, 2014. doi: 10.1127/0941-2948/2014/0551.

E. L. Fisher. An observational study of the sea breeze. *Journal of Meteorology*, 17(6):645–660, 1960. doi: 10.1175/1520-0469(1960)017<0645:AOSOTS>2.0.CO;2.

C. Hallgren, J. Arnqvist, S. Ivanell, H. Körnich, V. Vakkari, and E. Sahlée. Looking for an Offshore Low-Level Jet Champion among Recent Reanalyses: A Tight Race over the Baltic Sea. *Energies*, 13(14): 3670, 2020. doi: 10.3390/en13143670.

J. R. Holton. The diurnal boundary layer wind oscillation above sloping terrain. *Tellus*, 19(2):200–205, 1967. doi: 10.3402/tellusa.v19i2.9766.

P. C. Kalverla, J. B. Duncan Jr, G.-J. Steeneveld, and A. A. M. Holtslag. Low-level jets over the North Sea based on ERA5 and observations: together they do better. *Wind Energy Science*, 4(2):193–209, 2019. doi: 10.5194/wes-4-193-2019.

P. C. Kalverla, A. A. M. Holtslag, R. J. Ronda, and G.-J. Steeneveld. Quality of wind characteristics in recent wind atlases over the North Sea. *Quarterly Journal of the Royal Meteorological Society*, 146(728): 1498–1515, 2020. doi: 10.1002/qj.3748.

V. Kotroni and K. Lagouvardos. Low-level jet streams associated with atmospheric cold fronts: Seven case studies from the Fronts 87 Experiment. *Geophysical research letters*, 20(13):1371–1374, 1993. doi: 10.1029/93GL01701.

H. Rubio, M. Kühn, and J. Gottschall. Evaluation of low-level jets in the southern Baltic Sea: a comparison between ship-based lidar observational data and numerical models. *Wind Energy Science*, 7(6):2433–2455, 2022. doi: 10.5194/wes-7-2433-2022.

A.-S. Smedman, U. Högström, and H. Bergström. Low level jets – a decisive factor for off-shore wind energy siting in the Baltic Sea. *Wind Engineering*, pages 137–147, 1996. URL https://www.jstor.org/stable/43749611.

D. J. Stensrud. Importance of low-level jets to climate: A review. *Journal of Climate*, pages 1698–1711, 1996. URL http://www.jstor.org/stable/26201369.